# Effects of Different Structural Film Cooling on Cooling Performance in a GO$_2$/GH$_2$ Subscale Thrust Chamber

**Jixin Xiang [1,2], Yujie Jia [3], Zhiqiang Li [1,*] and He Ren [1,4]**

[1] College of Aeronautics and Astronautics, Taiyuan University of Technology, Taiyuan 030600, China; xiangjixin@tyut.edu.cn (J.X.); renhe@comac.cc (H.R.)

[2] Shanghai Aircraft Design and Research Institute, Shanghai 200436, China

[3] College of Mechanical and Vehicle Engineering, Taiyuan University of Technology, Taiyuan 030024, China; 15035575517@163.com

[4] Marketing Center, Commercial Aircraft Corporation of China Ltd., Shanghai 200126, China

[*] Correspondence: lizhiqiang@tyut.edu.cn

**Abstract:** To investigate the wall cooling of the thrust chamber in an engine, two film-cooling structures, namely, a circular hole structure and a slot structure, were designed. Numerical simulations were performed to study the coupled flow and regenerative cooling heat transfer in thrust chambers with different structures. The influences of parameters such as the film mass flow rate and film hole size on wall cooling were analyzed. Experiments were conducted in a thrust chamber to validate the accuracy of the numerical calculation method. The results indicate that the slot-structured film adheres better to the wall than the circular-hole-structured film, and the film closely adhering to the wall provides better insulation against hot gas, resulting in a reduction of approximately 6% in wall temperature. When the film hole size changes, the change in circumferential wall temperature in the upstream region of the slot-structured film is more pronounced. This paper aims to provide a reference for the design of the cooling structure at the head of the thrust chamber in engineering and suggests directions for optimization and improvement.

**Keywords:** liquid rocket engine; hydrogen–oxygen combustion; film cooling; numerical simulation

## 1. Introduction

In the past few decades, due to the high cost of rocket launches, many countries have focused their research on Reusable Launch Vehicles (RLVs) to significantly reduce launch expenses. As an ideal choice for executing space transportation missions, RLVs possess immense military and civilian value [1]. Additionally, they have garnered widespread attention both domestically and internationally in recent years due to their higher reliability, lower launch costs, and shorter turnaround times [2,3]. The reusable rocket engine is a critical component of the RLV, which is crucial for reducing rocket launch costs and enhancing launch reliability [4,5]. The rocket engine's combustion chamber operates in extremely harsh thermal conditions within a confined space, requiring effective thermal protection measures to ensure structural integrity at very high temperatures [6,7]. Further research has indicated that apart from the nozzle throat and injector regions, the thermal protection of the combustion chamber's head faces significant challenges because of the strong mixing and combustion of propellants near the injection face, resulting in complex thermal loads [8]. Studies have also suggested that temperature peaks often occur in the head region of a long cylindrical thrust chamber [9,10]. Hence, film cooling, as a crucial technique to protect the engine's combustion chamber head from direct exposure to hot gas surfaces, has gained widespread adoption [11].

Considerable research has been conducted in recent years on the performance of film cooling. Jonathan and Charles conducted numerical simulations on a small thrust

cryogenic/liquid oxygen engine equipped with head film cooling. They established computational models to simulate the combustion chamber's mixing and combustion processes, predicting the burning of the film and its protective capability on the wall surface [12]. Shine, S.R. Kumar, and others used a three-dimensional model for numerical simulations, studying the impact of factors such as blowing ratio, hole spacing, and hole diameter on film cooling characteristics. They identified an optimal blowing ratio for higher cooling efficiency within specific model geometry, offering reference for the analysis and optimization of straight-hole structures [13]. Arnold conducted numerical studies using a scaled combustion chamber, discussing important parameters affecting film-cooling efficiency and wall temperatures. They provided detailed insights into the influence of groove height and circumferential distribution on film-cooling efficiency [14]. Goldstein and others carried out numerous film cooling experiments early on, using air or helium for film cooling. They used the adiabatic wall temperature of helium and air secondary streams as dimensionless representations of film cooling efficiency, comparing experimental data with predictive results [15]. Michel and colleagues conducted experimental and numerical studies using a simplified combustion chamber platform with porous wall film cooling. This enhanced the understanding of film dynamics and established an experimental database on simplified geometries to validate numerical models [16]. S.R. Shine et al., conducted experimental research using air as the high-temperature gas and nitrogen as the coolant in a cylindrical test section to simulate the chamber's cooling effect. The results indicated that tangential injection reduced film cooling efficiency, while at high blowing ratios, far-field cooling effects were intensified [17]. A series of rocket engine thermal state tests were conducted, studying the influence of different parameters on film-cooling efficiency. Results showed higher temperatures near the inner wall close to the injection plate. Thus, the presence of film cooling significantly reduced the heat load on the chamber's inner wall. Moreover, the mass flow rate of the film had a significant impact on cooling efficiency, with increased film flow significantly enhancing cooling efficiency and providing more uniform circumferential cooling [18]. Currently, there has been extensive experimental and numerical analysis of the thrust chamber wall cooling regime. However, research on the cooling characteristics of different structured thrust chamber head film cooling holes and the effects of various parameters on the thrust chamber wall cooling regime remains quite limited.

This paper focuses on the thrust chamber and designs two types of head film cooling hole structures: a circular hole structure and a slot structure. Additionally, it includes the design of cooling channel structures to protect the thrust chamber cavity wall and acquire coupled temperature data. To validate the effectiveness of numerical simulation methods, relevant experiments were conducted on a combustion test platform to obtain experimental data. An EDC (Eddy-Dissipation Concept) model was employed to perform numerical simulations of oxygen/hydrogen combustion. The paper investigates the influence of parameters such as the proportion of film mass flow rate and the size of film-cooling holes on the wall surface-cooling results of circular-hole-structured film and slot-structured-film cooling in the thrust chamber. Furthermore, it analyzes the differences in wall surface cooling effects between the two structured configurations, elucidating the patterns of circular-hole-structured film and slot-structured film cooling on the thrust chamber walls. This research aims to provide a reference for the design of head cooling structures in engineering and offer optimization directions for its improvement.

## 2. Experiments Used for Validation

### 2.1. Experimental Apparatus

A set of compatible and reasonable pipeline supply systems has been established based on the existing conditions of the platform. The main propulsion supply system includes the oxygen supply module and the hydrogen supply module. Figure 1 illustrates the propulsion supply system, showing only the main components of the pipeline due to the complexity of the actual pipeline configuration. The auxiliary parts such as nitrogen operation and purging nitrogen are not presented in Figure 1. The gases are supplied

by large-capacity gas cylinders with a maximum supply pressure of up to 20 MPa. The gas flow is controlled by calibrated sonic nozzles, and the gas flow can be regulated by adjusting the pressure value in front of the sonic nozzle through a pressure regulator. In addition, the experiment includes the gas film supply module, with hydrogen serving as the gas film medium, which is supplied by a hydrogen cylinder. Considering the safety issues associated with the use of hydrogen and oxygen, thorough purging is conducted upstream of the hydrogen pipeline before each experiment. Additionally, after each ignition test, nitrogen is used to purge the downstream pipelines of both hydrogen and oxygen. Cooling water is supplied by a pump, and the mass flow rate of water can be adjusted during the experiment.

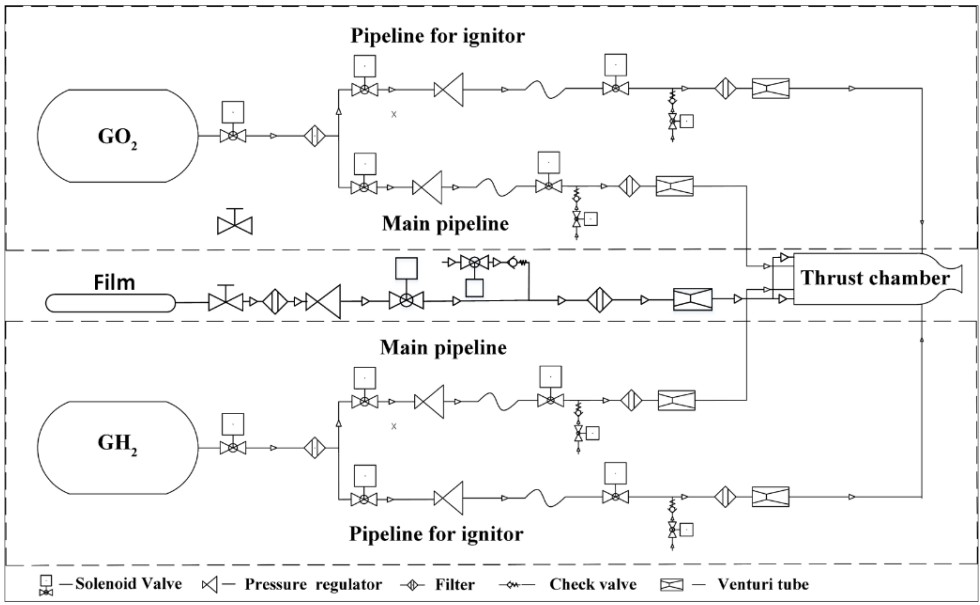

**Figure 1.** Schematic diagram of the test facility.

### 2.2. Thrust Chamber

The thrust chamber is composed of the combustion chamber head, combustion chamber cylindrical segment, igniter, and nozzle segment. The combustion chamber head mainly consists of the oxygen head cavity, hydrogen head cavity, injector, and air film supply segment. Fuel and the oxidizer are mixed and burned in the cylindrical segment, and the combustion gases are accelerated through the nozzle before being ejected from the outlet. The air film injection segment, due to its lower temperature, is made of stainless steel. The cylindrical segment is made of high thermal conductivity purple copper material, which is similar to the actual engine's inner wall material. The structure of the thrust chamber is shown in Figure 2. The specific dimensions of the thrust chamber are provided in Table 1.

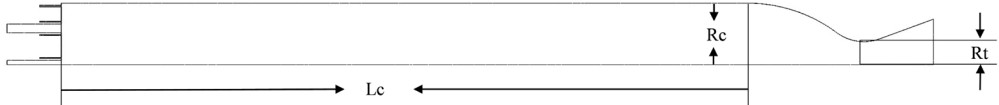

**Figure 2.** Structure of the thrust chamber.

**Table 1.** Geometry parameters of the combustion chamber.

| Parameter | Value (mm) |
|---|---|
| $L_c$ | 380.0 |
| $R_c$ | 33.9 |
| $R_t$ | 12.7 |

The oxygen head cavity in the combustion chamber head serves as an interface for oxygen introduction and allows oxygen to gather in the cavity. The hydrogen head cavity is directly connected to the oxygen head cavity. A coaxial shear coaxial gas–gas injector inside the hydrogen head cavity is used to organize hydrogen and oxygen, which are sheared out from 7 coaxial nozzles for mixing combustion. The schematic diagram of a single nozzle gas–gas injector is shown in Figure 3. The design criteria for the gas–gas injector refer to the book "Liquid Rocket Engine Gas–Gas Combustion and Gas–Gas Injector Technology". The dimensions of the 7 injectors are the same, and specific values are provided in the table. The nozzles are distributed in two rings, with 1 nozzle in the inner ring and 6 nozzles evenly distributed in the outer ring. The layout schematic of the nozzles is given in Figure 4. The nozzle dimensions are provided in Table 2.

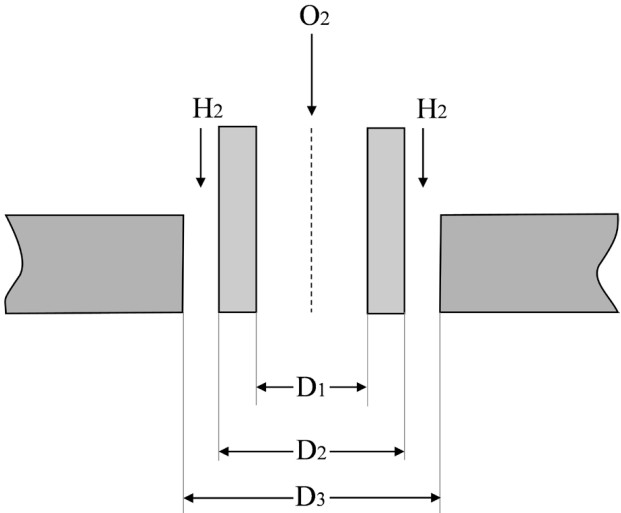

**Figure 3.** Schematic diagram of a single nozzle gas-to-gas injection.

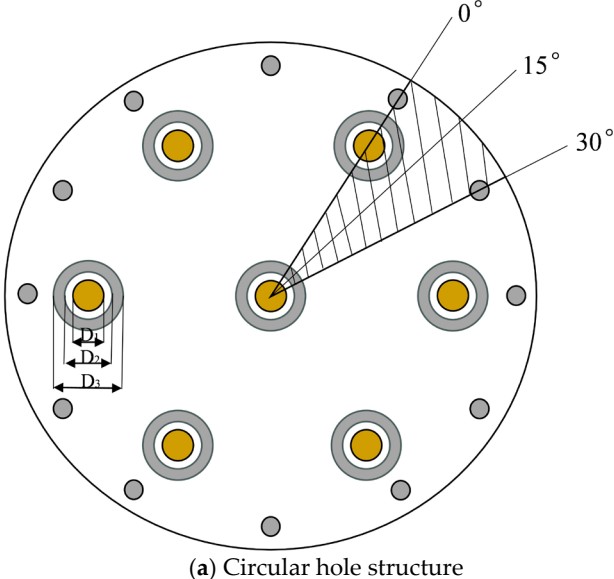

(**a**) Circular hole structure

**Figure 4.** *Cont.*

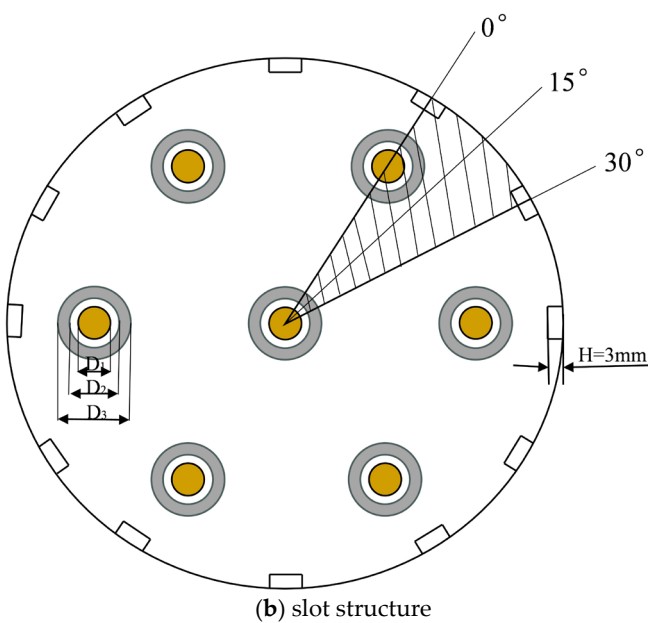

(**b**) slot structure

**Figure 4.** Nozzle layout of the injection panel.

**Table 2.** Parameters of the nozzle.

| Parameter | Value (mm) |
|:---:|:---:|
| $D_1$ | 4.8 |
| $D_2$ | 6.8 |
| $D_3$ | 7.6 |

*2.3. Experimental Analysis*

As shown in Figure 4, the test specimen exhibits symmetry in the circumferential direction. Therefore, it is possible to install three rows of thermocouples in the thrust chamber at intervals of 30°, with each row spaced 15° apart, capturing a minimum cycle of 30°. The reference points for these rows are defined, with 0° corresponding to the downstream wall position directly facing the outermost nozzle in the multi-nozzle injection panel. Position 30° corresponds to the downstream wall position between the two outer nozzles, and position 15° is located between the two aforementioned positions. Along the axial direction, three sets of measurement points are evenly distributed at eight different axial positions. Custom-designed sheathed K-type coaxial thermocouples are utilized for measurements. The temperature range for measurement was limited to 0~900 °C. Each row of thermocouples is distributed axially at seven different locations. Due to the temperature changes being more pronounced closer to the nozzle panel, and by the principle of placing the thermocouples upstream in a dense arrangement and downstream in a sparse arrangement to capture more thermal flow variation information, while also considering manufacturing issues, the spacing between the two should not be too close. Therefore, the spacing method described in the text is adopted, with more details shown in Figure 5, and the distances of the thermocouples relative to the panel are specified in Table 3.

**Table 3.** Thermocouple distances relative to the panel.

| Number | P1 | P2 | P3 | P4 | P5 | P6 | P7 | P8 |
|:---:|:---:|:---:|:---:|:---:|:---:|:---:|:---:|:---:|
| Z [mm] | 3.0 | 18.0 | 33.0 | 53.0 | 81.0 | 109.0 | 139.0 | 169.0 |

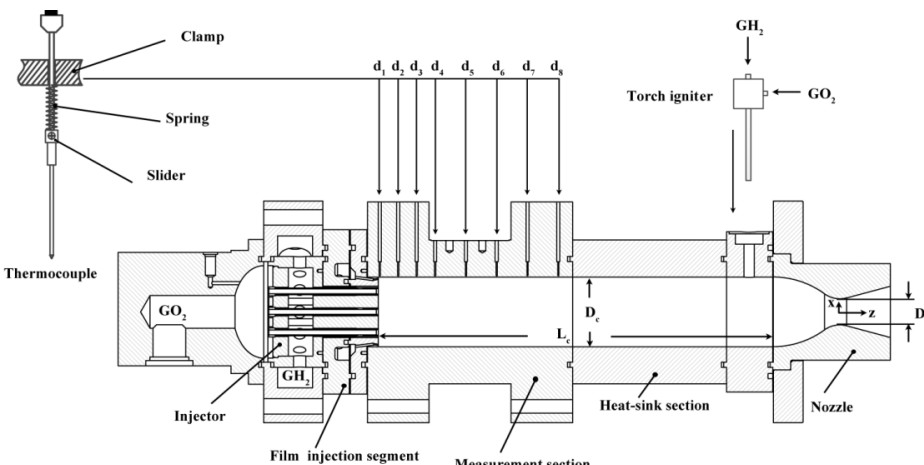

**Figure 5.** Thermocouple position.

The film supply segment is situated between the hydrogen head cavity and the cylindrical segment. During the experiment, the film medium was injected along the outer edge of the injector, forming a film-cooling layer to protect the wall surface. The film supply section consists of the main section and the exit section, forming a film cavity to ensure uniform flow of the film before entering the nozzle. The main section is upstream and introduces the film medium through four radially distributed pipelines. The exit section is sealed to the main section with graphite gaskets, and the desired film distribution is achieved at the exit end through the design of the exit structure.

In film cooling, the mass flow rate of the film is a major factor influencing cooling efficiency. Generally, cooling effectiveness increases with an increase in film mass flow rate. To achieve better cooling effects, the film mass flow rate should be increased as much as possible, provided it does not adversely affect the combustion process. However, in rocket engines, the film cooling medium is typically one of the propellants, and introducing film cooling can affect the local mixture ratio, altering the combustion process and, consequently, the engine's combustion performance. Therefore, in the study of head film cooling, in addition to investigating the influence of different film parameters on cooling efficiency, changes in combustion performance must also be considered.

In the experimental conditions, the film hole diameter was 1.2 mm, the film mass flow rate ratio was around 10%, and the mixture ratio of the propellants was approximately 5.8. To ensure steady-state combustion, the duration of all conditions in the hot test was 3 s. Data from thermocouples and pressure sensors were smoothed using the Savitzky–Golay filter function to eliminate noise caused by interference [19]. Error analysis of the measurement data showed that the measurement error of heat flux is less than $\pm1.4\%$, and the temperature measurement error was $\pm0.4\%$. Therefore, the uncertainty of measuring the wall temperature can be largely ignored in its impact on heat flux [20]. During the experimental measurement process, the radii of the inner and outer walls of the combustion chamber were measured using a vernier caliper with a measurement accuracy of $\pm0.02$ mm. The depth of the combustion chamber measurement orifices was measured with an accuracy of $\pm0.1$ mm. The distance from the measurement point to the inner wall of the combustion chamber was measured with an accuracy of approximately $\pm0.104$ mm.

The propellant mass flow rate was calculated using the following formula:

$$\dot{m}_g = \frac{p_c A_t}{c^*} \tag{1}$$

where Pc is the combustion chamber pressure, At is the throat cross-sectional area of the nozzle, and C* is the characteristic velocity. In each condition, the flow rates of oxidizer and fuel can be calculated using the following formulas:

$$\dot{m}_{H_2} = \frac{1}{MR+1}\dot{m}_g \tag{2}$$

$$\dot{m}_{O_2} = \frac{MR}{MR+1}\dot{m}_g \tag{3}$$

where MR is the propellant mixing ratio, which is the ratio of the oxidizer mass flow rate to the fuel mass flow rate. The MR in the text is a fixed value, which is typically around 5.9. The combustion efficiency of each test is calculated as

$$\eta_c = \frac{p_{c,\exp}}{p_{c,th}} \tag{4}$$

where $P_{c,\exp}$ and $P_{c,th}$ represent the experimental combustion chamber pressure and the theoretical combustion chamber pressure calculated from thermodynamics, respectively.

## 3. Grid Generation and Boundary Conditions

The computational domain comprises the film/hot-gas domain and the coolant/cooling channel domain. To reduce computational costs, these two computational domains are handled separately. On the coupling surface of the coolant/cooling channel, a no-slip constant heat flux boundary condition is set, and the walls on both sides of these two domains are symmetrical boundaries. Using "Liquid Rocket Engine Design" as a reference ensures that the wall remains within reasonable operating temperatures. The coolant mass flow rate was designed at 2.4 kg/s, with an inlet temperature of 300 K for the coolant and a cooling channel height of 2 mm. The inner and outer walls of the thrust chamber were made of copper, with a thickness of 2 mm and 1 mm, respectively.

The film/hot-gas domain is divided into upstream and downstream regions. In the upstream region, there is intense combustion among propellants, making it a significant area influenced by the film. Therefore, the grid nodes are densely arranged. In contrast, the downstream region witnesses more extensive gas and film flow, with diminished film influence, resulting in sparser grid node arrangements. These two grid sections are connected through an interface. The walls on both sides of these two regions are symmetric boundaries, and a no-slip constant temperature boundary condition is defined on the coupling surface. Due to the propulsion chamber being filled with low-speed flowing hydrogen on the other side of the injector plate, the heat flow transferred from the combustion chamber to the injector plate significantly surpasses the convective heat transfer away from the plate within the chamber. Hence, the injector plate can be set as an adiabatic boundary condition. Additionally, all faces not specified in the figure are set as no-slip adiabatic boundary conditions.

The film/hot-gas domain was generated in Gambit 2.4.6 software, as shown in Figure 6. Mostly, structured hexahedral grids were used in this area. Compared to the downstream area, the upstream area has a denser distribution of grid nodes, capturing flame structures and film cooling characteristics accurately. Circular-hole-structured film combustion chambers and slot-structured film thrust chambers differ only in their head regions. The grid for the coolant/cooing channel domain was generated in ICEM CFD 2022R1 software and was entirely divided using structured hexahedral grids due to its simple geometric structure, as shown in Figure 7. On the coupling surfaces of all different regions, the grid nodes between the two different regions do not match, but the total number of high-density grid nodes on the coupling surfaces does not exceed 20 times the total number of low-density region coupling surface nodes.

Due to the geometric model's circumferential symmetry concerning the boundary conditions, the 1/12th part of the overall model serves as the computational domain for this study to reduce the computational load. The inlet boundary conditions for the computational model are labeled in the figure, all using mass flow rate inlet conditions. The nozzle exit uses a pressure boundary with a pressure value of atmospheric pressure, 0.101325 MPa. The outlet for external coolant also uses a pressure boundary with a pressure

value of 1.50 MPa, which is the design pressure for the thrust chamber. The outer wall of the thrust chamber is subjected to an adiabatic boundary condition, and the boundaries perpendicular to the flow direction for the gas region, external coolant region, and thrust chamber wall are set as symmetrical boundary conditions. The gas film's inlet mass flow rate varies based on the calculated operating conditions.

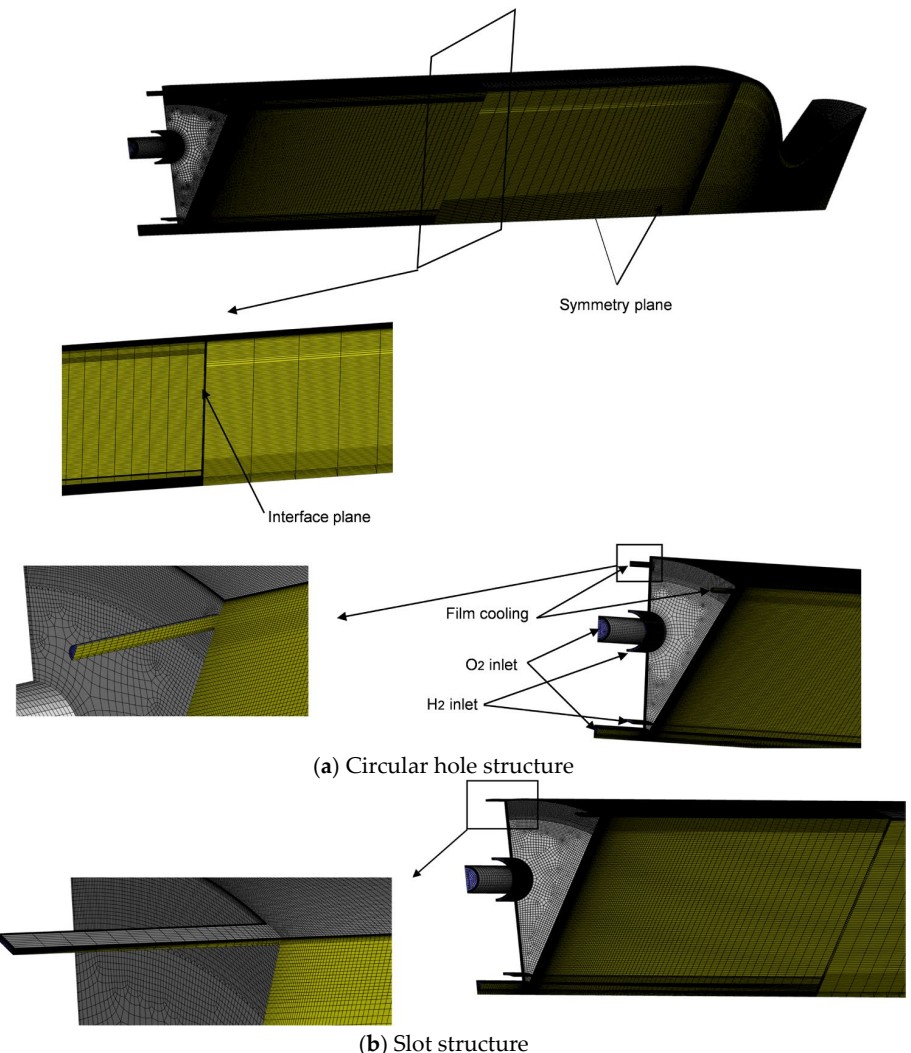

(**a**) Circular hole structure

(**b**) Slot structure

**Figure 6.** Schematic diagram of the film/hot-gas region mesh.

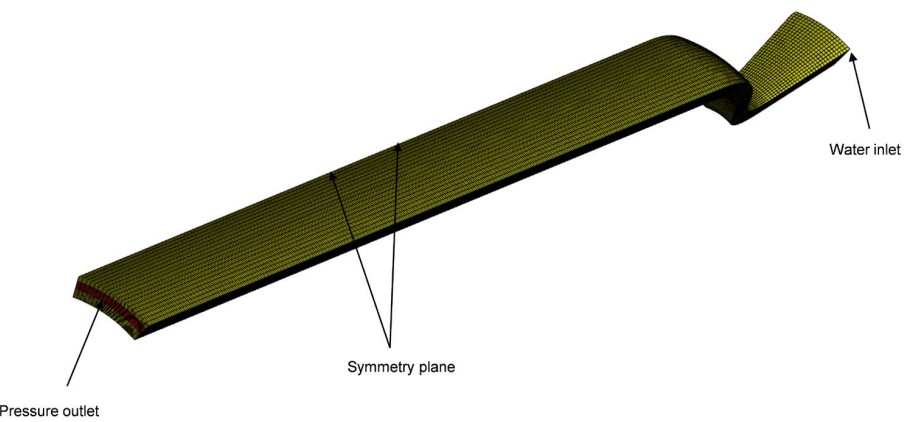

**Figure 7.** Schematic diagram of the coolant/cooling channel region mesh.

### 4. Numerical Analysis

In the calculation of hydrogen–oxygen combustion, if the complete hydrogen–oxygen diffusion combustion process is considered, it is necessary to include a large number of fuel molecules, oxidizer molecules, and various other intermediate species participating in chemical reactions. To reduce the impact of chemical reaction mechanisms on the calculation results of the combustion chamber and improve computational efficiency, several types of mechanisms were analyzed and compared. The hydrogen–oxygen chemical reaction process was reduced to six components and nine steps of chemical reactions [21]. The specific parameters of the chemical kinetic model are shown in Table 4. The study based on engineering research on the effects of different film-cooling holes, disregarded the influence of hole positioning. For a detailed examination of the positional effects, refer to Appendix A.

**Table 4.** The hydrogen–oxygen six-species nine-step chemical reaction kinetic model.

| Serial Number | Reaction Equation | $A_f$ | $\beta_f$ | $E_f$ [J/kmol] |
|:---:|:---:|:---:|:---:|:---:|
| 1 | $H_2 + O_2 = 2OH$ | $1.7 \times 10^{13}$ | 0 | 200 |
| 2 | $H_2 + OH = H_2O + H$ | $1.17 \times 10^9$ | 1.3 | 15.17 |
| 3 | $2OH = H_2O + O$ | $5.9 \times 10^9$ | 1.3 | 71.25 |
| 4 | $H_2 + O = OH + O$ | $1.8 \times 10^{10}$ | 1.0 | 36.93 |
| 5 | $O_2 + H = OH + O$ | $2 \times 10^{14}$ | 0 | 70.3 |
| 6 | $H + O + M = OH + M$ | $6.0 \times 10^{16}$ | −0.6 | 0 |
| 7 | $2O + M = O_2 + M$ | $6.17 \times 10^{15}$ | −0.5 | 0 |
| 8 | $2H + M = H_2 + M$ | $1.8 \times 10^{18}$ | −1.0 | 0 |
| 9 | $OH + H + M = H_2O + M$ | $1.17 \times 10^9$ | −2.0 | 0 |

Note: M denotes body 3.

Where $A_f$ is the pre-exponential factor; $\beta_f$ is the temperature factor; $E_f$ is the activation energy of the reaction. The forward chemical reaction rate of reaction r is calculated by the Arrhenius formula:

$$k_{f,r} = A_f T^{\beta_f} e^{-E_f/RT} \tag{5}$$

In the calculations in this paper, the interaction between turbulence and chemical reactions was calculated using the Eddy Dissipation Concept (EDC) model [22]. The Eddy Dissipation Concept model is an extension of the Eddy Dissipation model, and the EDC model can include a more detailed chemical reaction mechanism. It is assumed in the calculation that chemical reactions occur at a small scale (small turbulent structures). The source term of the continuous equation for component i is represented by Equation (6):

$$R_i = \frac{\rho(\xi^*)^2}{\tau^* \left[1 - (\xi^*)^3\right]}(Y_i^* - Y_i) \tag{6}$$

where * represents the small scale quantity, and $\xi^*$ is the length ratio of the small scale, which can be calculated by Equation (7):

$$\xi^* = C_\xi \left(\frac{v\varepsilon}{k^2}\right)^{1/4} \tag{7}$$

where $\tau^*$ is the time scale for the initiation of the reaction, controlled by the Arrhenius reaction rate, and can be calculated by Equation (8):

$$\tau^* = C_\tau \left(\frac{v}{\varepsilon}\right)^{1/2} \tag{8}$$

The numerical simulations in this paper were mainly conducted on the Fluent 2022R1 software platform. This software platform, as an important general combustion flow field calculation tool in the field of computational fluid dynamics, can effectively handle

flow field calculation problems in continuous flow. Fluent primarily solves the coupled Reynolds time-averaged N-S equations through discrete grids [23]. The control equation set was discretized using the finite volume method, where the convective term adopted a second-order upwind format, and the diffusion term adopted a central difference format. The calculation used the Fluent pressure-based solver and the $k - \omega$ turbulence model. The coupling between pressure and velocity was handled using the coupled algorithm. Convergence was considered achieved when the residuals of each discrete term reached within $10^{-4}$ and the computational parameters stabilized. In the simulation process, for $GH_2$ and $GO_2$, the specific heat (Cp) was calculated using piecewise polynomial, and the viscosity and thermal conductivity were determined using the kinetic theory.

The simulation employs an EDC model, embedding the thermal–physical properties of the propellant and combustion gas into the solver through User-Defined Functions (UDFs). During computation, Fluent directly invokes UDFs for calculations. Assuming the temperature distribution on the gas side wall, we calculated the corresponding assumed heat flux density and used the assumed wall temperature distribution as boundary conditions for simulating the film/hot-gas region to obtain the coupled surface heat flux density distribution. Simultaneously, we used the assumed heat flux density as boundary conditions for simulating the coolant/cooling channel region to obtain the coupled surface temperature distribution.

By averaging the computed wall temperature and assumed wall temperature as the boundary conditions on the coupled surface, we performed another simulation of the film/hot-gas region to obtain the distribution of new heat flux density on the coupled surface. Similarly, we averaged the computed heat flux density and assumed heat flux density as the boundary conditions on the coupled surface; then, we performed another calculation of the coolant/cooling channel region to obtain the new wall temperature on the coupled surface. We repeated this process until the relative error of heat flux density and wall temperature for two consecutive iterations did not exceed 1%. The specific process is shown in Figure 8.

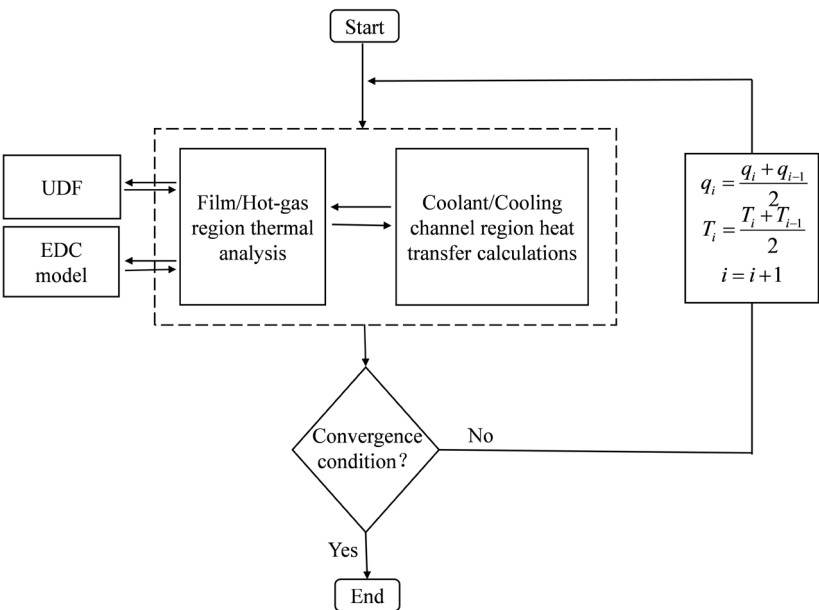

**Figure 8.** Schematic diagram of the process.

## 5. Grid Independent Test

To ensure the mesh independence of the numerical simulation, a mesh independence analysis was conducted for a specific operating condition. Three different grid resolutions were calculated with grid numbers of 0.65 million, 1.4 million, and 2.1 million, respectively. Figure 9 compares the results of gas-side heat flux calculated using different grid resolutions. From the figure, it can be observed that upstream and downstream of the combustion

chamber, the heat flux calculated by grids of case 2 and case 3 are almost consistent, while the wall temperature calculated by case 1 exhibits a larger error. In the middle section of the combustion chamber, the wall temperature differences obtained from the three types of grids are relatively small. Therefore, considering the overall situation of the combustion chamber calculations and aiming to reduce computational costs, case 2 grids can accurately and effectively calculate the flow and heat transfer in the film-cooled thrust chamber. Consequently, this resolution of the grid is adopted in subsequent simulations.

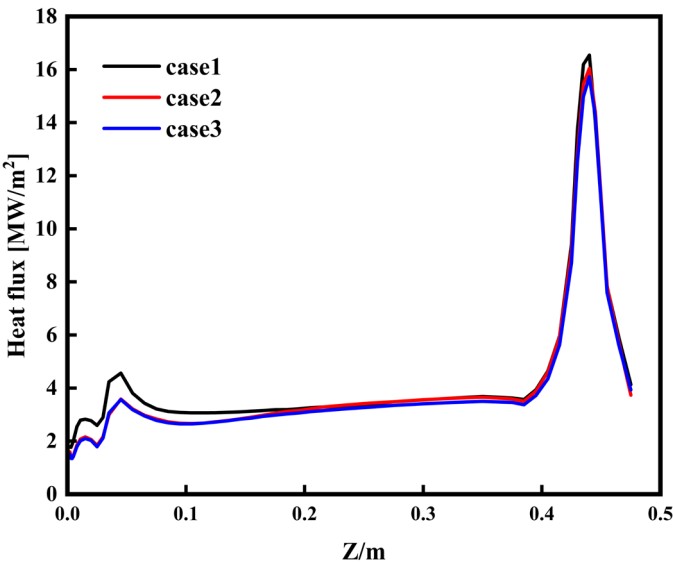

**Figure 9.** Hot-gas-side wall temperatures for various grid types.

## 6. Results and Discussion

### 6.1. Experimental Results and Validation

In the experiment, the temperature inside the wall was measured using thermocouples, and then the gas-side wall temperature and heat flux were obtained by solving the heat conduction inverse problem based on single-point temperature measurement. In the calculation process, it is assumed that the gas-side wall is under a constant heat flux boundary condition. Initially, the heat flux was iteratively adjusted to obtain the average heat flux. Subsequently, based on the results of the average heat flux, transient heat flux was determined. Using the obtained transient heat flux as the constant heat flux boundary condition on the gas-side wall, the transient temperature of the inner wall could be calculated. This method allows for the derivation of the corresponding heat flux based on known non-steady-state wall temperatures [24]. To confirm the accuracy of the simulation model, experimental conditions were set with a membrane hole diameter of 1.2 mm, a gas film flow mass rate ratio of around 7.5% and 12.5%, and a propellant mixture ratio of approximately 5.8. In the experiments, to ensure steady-state combustion, the duration of all test conditions was 3 s. When validating the model, the experimentally obtained gas-side wall temperature distribution was applied as the first type of boundary condition on the combustion chamber wall, and the corresponding heat flux was then calculated. The feasibility of the numerical simulation was verified by comparing the simulated heat flux with the experimental data.

Figure 10 shows a comparison between the numerically simulated heat flux and experimental data. Additionally, both the calculated wall heat flux curve and experimental data exhibit a local maximum at the axial position Z = 0.04 m due to flame expansion with the numerically simulated local maximum slightly exceeding the experimental data. This discrepancy may be attributed to the simplified chemical reaction mechanism used, which overestimates the completion degree of propellant combustion in this region. At Z = 0.03 m, under the condition where hydrogen accounts for 12.5%, the calculated wall

heat flux density differs from the experimental value by 0.45 MW with a relative error of approximately 10%. This level of deviation is acceptable in engineering applications, indicating that the computational model can be used for subsequent research.

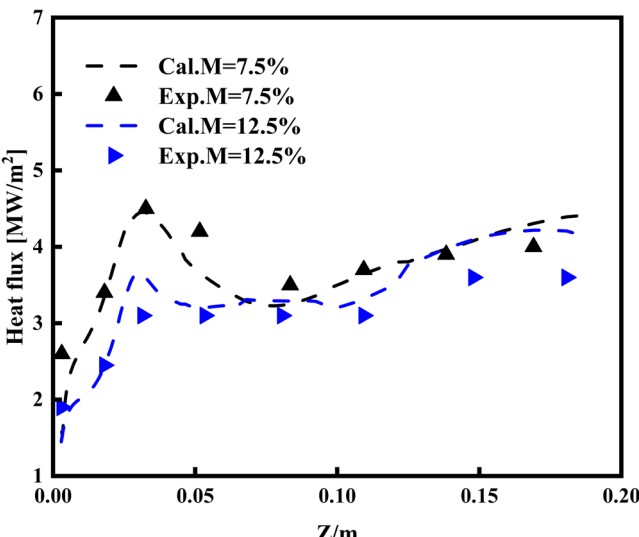

**Figure 10.** Comparison of experimental and calculated axial profiles of heat flux.

*6.2. Influence of Mass Flow Rates*

To investigate the impact of mass flow rate on the cooling performance of the circular-hole structure and slot structure film cooling in a thrust chamber, different operating conditions were considered with film mass fractions of 3.6%, 6.5%, and 10% hydrogen. Table 5 shows the mass flow rates of the propellant and the gas film for different gas film flow rate ratios. Figure 11 illustrates the variation in circumferentially averaged gas-side wall temperatures with axial distance for circular-hole-structured film and slot-structured film at different mass flow rates. It is observed that as the mass flow rate increased, both the circular-hole-structured film and slot-structured film exhibited improved cooling effects in the wall of the combustion chamber. This improvement is attributed to the increased hydrogen near the wall at higher mass flow rates, enhancing thermal protection and resulting in a significant reduction in wall temperatures. Comparing the two structures at the same mass flow rate, the slot-structured film tends to achieve lower cooling temperatures in the upstream region of the combustion chamber, with the maximum temperature decreasing by approximately 6%. Figure 12 depicts the streamlines of gas flow in the region, while Figure 13 shows the temperature contour map upstream of the combustion chamber. From the figure, it can be observed that on one hand, the introduction of the coolant generates a low-pressure zone due to the step effect. On the other hand, due to the compression of the flame nodes formed by the main gas flow, there is a phenomenon of coolant backflow in the combustion chamber. The coolant initially flows due to the influence of initial velocity, but after reaching a certain distance, recirculation of the coolant occurs, forming vortices. These vortices tend to carry away the coolant near the wall, resulting in reduced cooling efficiency. The circular-hole-structured film generates noticeable vortices at the hole exit, which displace the coolant away from the wall, diminishing the cooling efficiency. In contrast, the coolant from the slot-shaped structured film-cooling is entrained downstream by the high-temperature mainstream gas. Therefore, in the upstream region of the combustion chamber, the slot-structured film achieves lower wall temperatures.

**Table 5.** The gas film mass flow rate parameter.

| $f_{film}$ | $\dot{m}_g[g/s]$ | $\dot{m}_{film}$ |
|---|---|---|
| 10.0% | 288 | 4.8 |
| 6.5% | 305 | 3.4 |
| 3.6% | 320 | 2.0 |

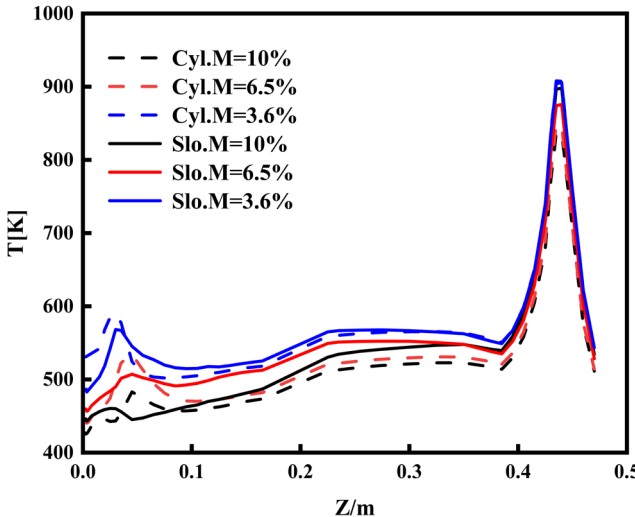

**Figure 11.** Comparison of wall temperatures for various mass flow rates.

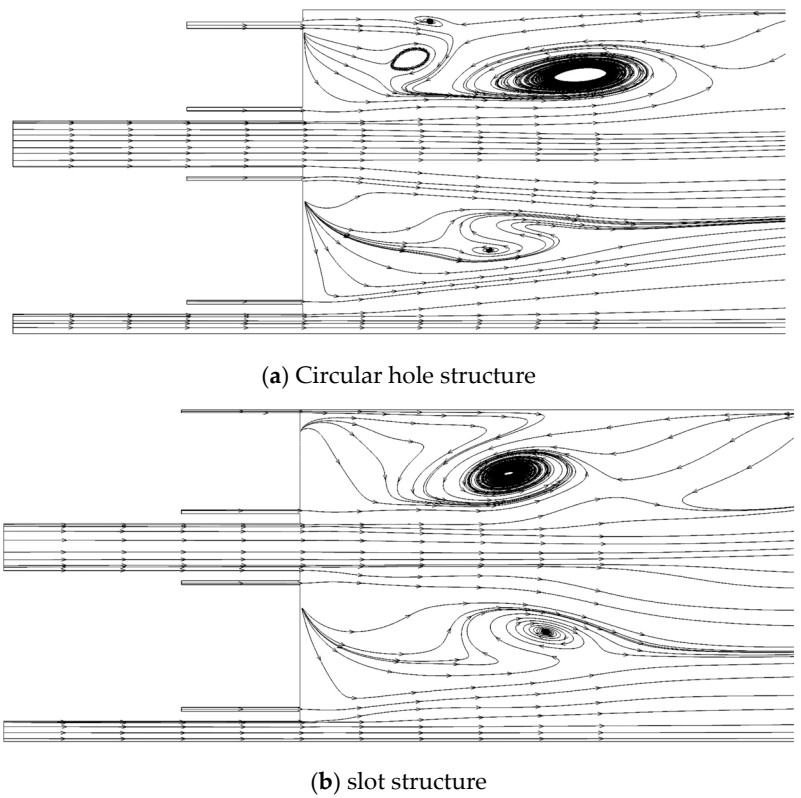

(**a**) Circular hole structure

(**b**) slot structure

**Figure 12.** Comparison of film streamlines for two structures.

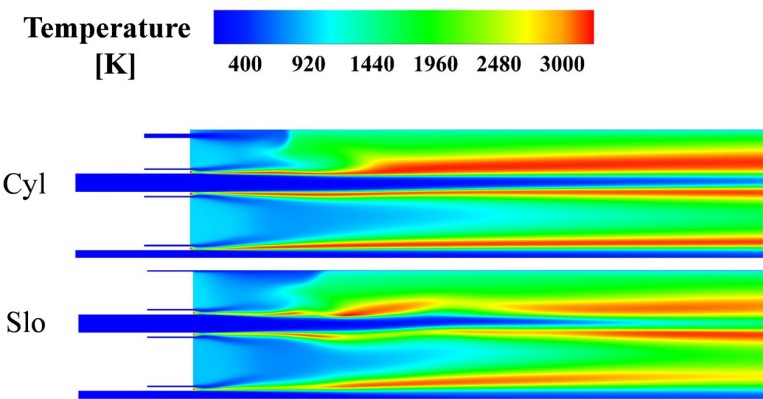

**Figure 13.** Temperature distribution map upstream of the combustion chamber.

Figure 14 shows the variation in the highest wall temperature in the upstream region of the combustion chamber with mass flow rate for both circular-hole-structured film and slot-structured film. It is evident that as the mass flow rate increases, the highest wall temperatures for both structures consistently decrease. This reduction is attributed to the increased coverage of the wall by the cooling gas as the mass flow rate rises, resulting in a continuous decrease in the highest wall temperatures. Additionally, with the decrease in air film mass flow, the location of the highest wall temperature is closer to the injection panel. Figure 15 displays streamlined plots for the circular-hole-structured film at different mass flow rates. As the mass flow rate decreases, gas recirculation occurs under the pressure differential, filling the downstream region below the nozzle with enlarging vortices. Simultaneously, as the coolant mass decreases, the film length shortens, causing the vortices formed by the coolant recirculation to diminish and gradually approach the panel. Consequently, with decreasing mass flow rates, the location of the highest wall temperature moves closer to the injection panel.

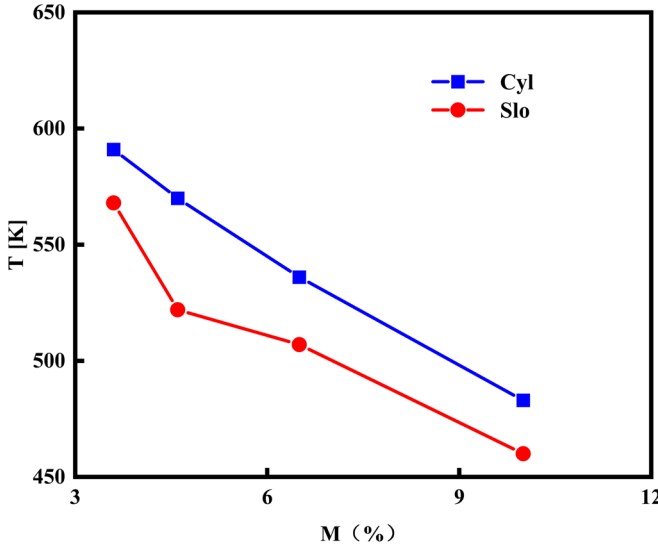

**Figure 14.** Schematic diagram of temperature peak variation with mass flow rates.

These results indicate that under the same mass flow rate conditions, each structure has its advantages in terms of cooling effectiveness in the combustion chamber. Therefore, the choice between the circular hole structure and slot structure in the design of combustion chamber film cooling should consider both the mass flow rate and specific cooling locations. Moreover, considering the processing difficulty of film cooling hole structures, circular-hole-structured films are easier to manufacture.

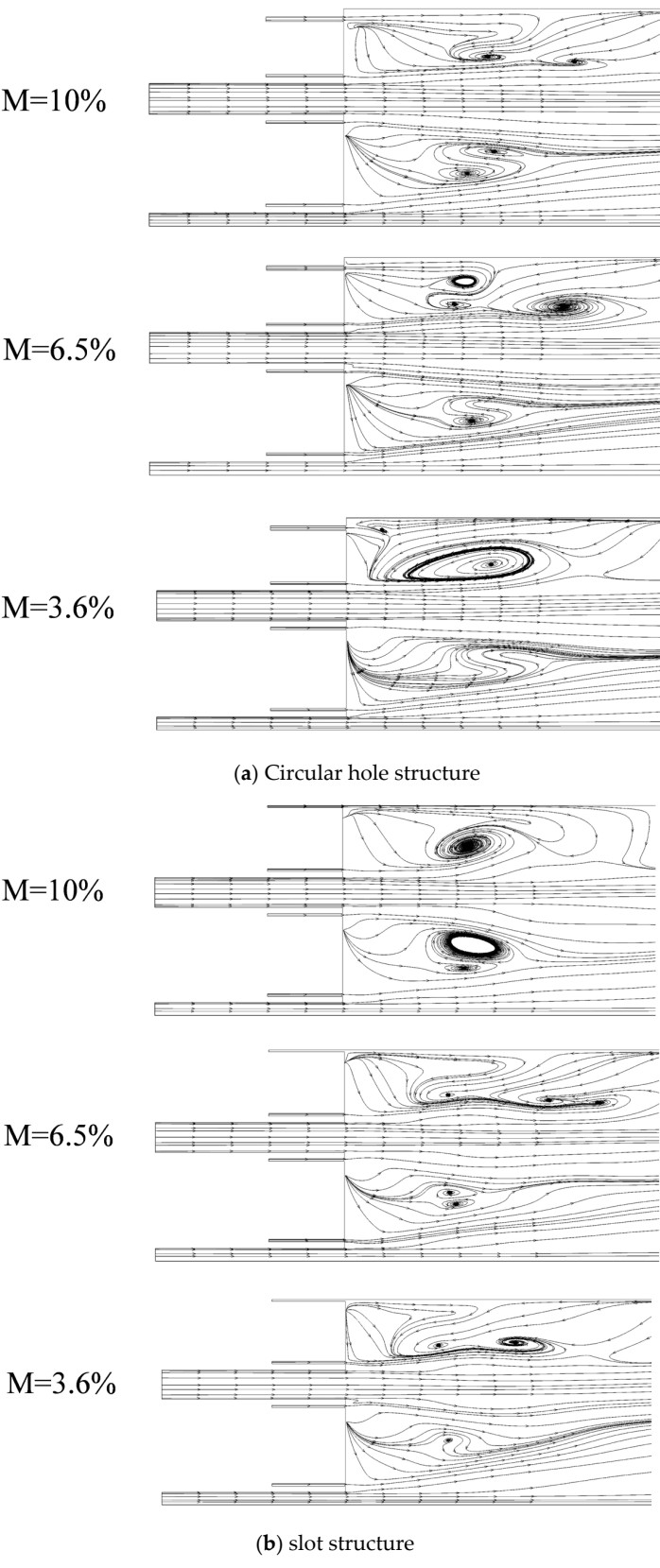

(**a**) Circular hole structure

(**b**) slot structure

**Figure 15.** Film streamlines with mass flow rates.

### 6.3. Influence of Film Hole Size

To investigate the influence of air film hole size on the cooling performance of circular-hole-structured film and slot-structured film in the thrust chamber, designs were made for

a circular-hole-structured film and slot-structured film with the same hole cross-sectional area. The diameters of the circular-hole-structured film were 1.2 mm, 1.0 mm, and 0.8 mm, corresponding to areas of 1.13 mm², 0.78 mm², and 0.50 mm², respectively. In comparison, the slot-structured film had a hole thickness of 0.3 mm and widths of 6.37°, 4.42°, and 2.83°.

Figure 16 depicts the axial variation in the circumferential average gas-side wall temperature for circular-hole-structured film and slot-structured film with different air film hole sizes. From the figure, it can be observed that for the circular-hole-structured film, as the air film hole size increased, the wall temperature in the region near the panel increased. However, as the distance from the panel increased, the temperature difference became negligible. In the downstream region of the combustion chamber, higher wall temperatures are observed for conditions with larger air film hole sizes. This is because near the exit of the air film holes, with a constant air film flow rate, smaller hole diameters lead to lower disturbance of combustion gas in the thrust chamber, allowing the air film to better protect the thrust chamber wall and ensure higher cooling efficiency. When the film hole diameter is 0.8 mm, 1.0 mm, and 1.2 mm, the average velocity of the cooling film inlet in the slot structure is 596 m/s, 298 m/s, and 156 m/s, and the average velocity of the circular hole structure is 619 m/s, 306 m/s, and 160 m/s, respectively. When the size of the air film pore is larger, the flow rate of the air film coolant is lower, and the initial momentum is smaller, so the coolant has a shorter film protection length, which in turn affects the protection of the wall. Consequently, in the downstream region, as the hole diameter of the gas film increases, the wall temperature also becomes higher.

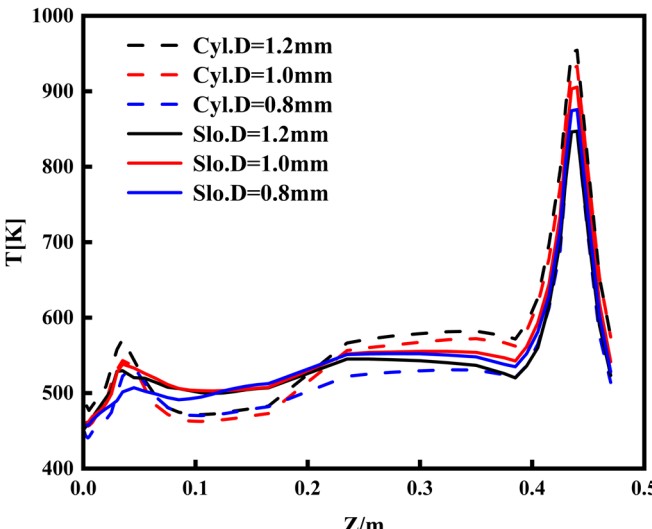

**Figure 16.** Comparison of wall temperatures for various film hole size.

In the upstream region of the combustion chamber, the wall temperature of the slot-structured film is significantly influenced by the size of the film holes. As the size increases, the wall temperature rises. However, as the size continues to increase, the temperature no longer rises but instead stabilizes. However, as the distance from the panel increases, the influence of air film hole size on wall temperature gradually diminishes, and the temperature difference becomes small. Figure 17 presents the variation in the $H_2$ mole fraction along the gas flow direction for the circular-hole-structured film and slot-structured film at different sizes. From the figure, it can be seen that the variation in the $H_2$ mole fraction is more pronounced for the slot-structured film with significant decreases in short distances. When the air film hole size changes, the cooling medium on the combustion chamber wall decreases rapidly, causing the wall temperature to be affected by the air film hole size in advance. Therefore, as the distance from the panel increases, the wall temperature difference becomes negligible in the downstream region of the combustion chamber.

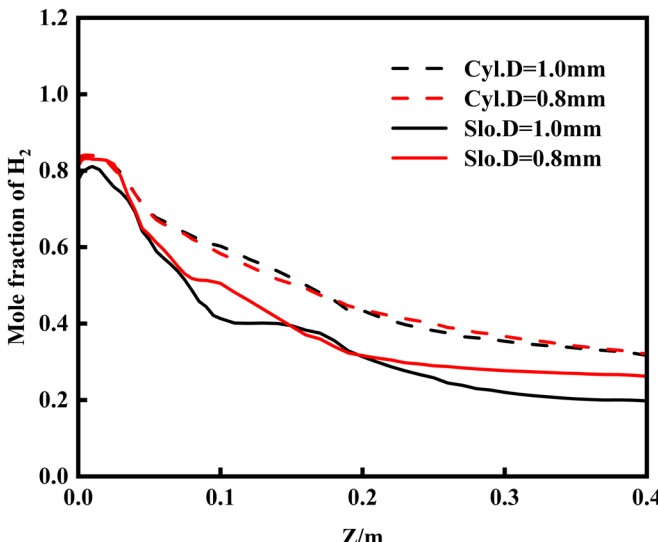

**Figure 17.** Flow direction variations of the $H_2$ mole fraction.

The decrease in the $H_2$ mole fraction results in the regeneration coolant absorbing more heat, increasing the circumferential uneven change caused by the cooling channel. Hence, the circumferential variation in the slot-structured film is more pronounced. This is also because the slot-structured film holes are close to the wall, and changes in air film hole size only affect circumferential changes, making the impact on circumferential wall temperature more apparent. In order to explain the reason behind the variation in the upstream wall temperature of the slot-structured film combustion chamber with changes in film hole size, Figure 18 provides the results of the surface heat flux in polar coordinates for different sizes. Due to the symmetry of the injector structure, the heat flux values vary periodically with a 60-degree cycle in the circumferential direction. Numerical analysis is performed on cross-sections at distances of 21 mm, 31 mm, 41 mm, and 61 mm from the injection panel along the axial direction.

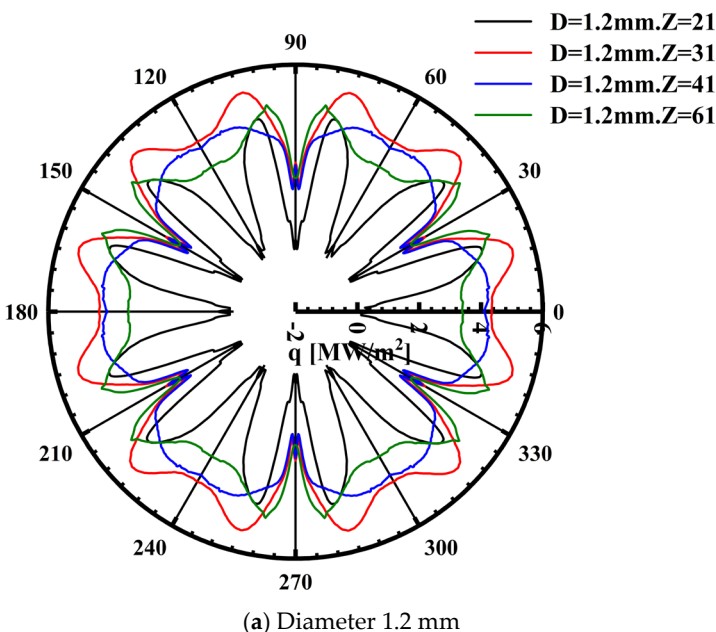

(**a**) Diameter 1.2 mm

**Figure 18.** *Cont.*

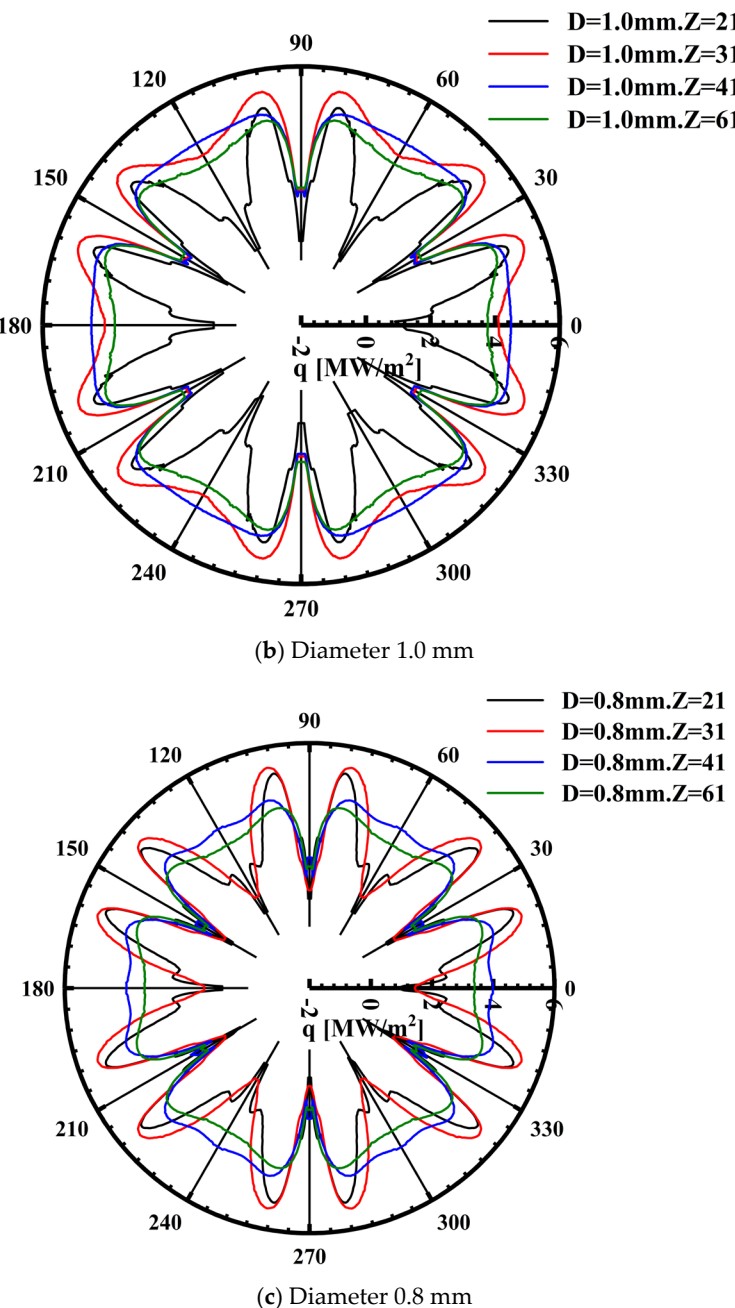

(**b**) Diameter 1.0 mm

(**c**) Diameter 0.8 mm

**Figure 18.** Circumferential variations of wall heat flux at four axial positions.

From the figure, it can be observed that for the same air film hole size, the region near the injection panel exhibits minimum heat flux values at 0°, 30°, and 60°, while maximum values occur between 0° and 30° and between 30° and 60°. As the axial distance from the panel gradually increases, minimum heat flux values only appear at 30°, while the values at other positions remain relatively high. This is because cooling air film holes appear at 0°, 30°, and 60°. The hydrogen/oxygen injection exits at 0° and 60°, where hydrogen and oxygen mix and combust, generating a large amount of heat. In the region near the injection panel, the cooling effect of air film holes results in lower heat flux values at 0°, 30°, and 60°. However, in the position between the two air film holes, insufficient coverage of cooling gas leads to higher heat flux values. As the distance from the panel gradually increases, the effect of air film holes weakens, and hydrogen and oxygen gradually mix completely and burn, causing heat flux values at 0° and 60°, located at the exits of the hydrogen/oxygen injection, to rise. With an increase in air film hole size, the positions

of high heat flux at 0° and 60° also move closer to the panel. This is because larger air film hole sizes result in smaller hole injection capacities, and the axial distance affected is shorter. Next, we consider the Z = 31 mm curve, which is the curve in Figure 16 at the location of the maximum temperature in the upstream region of the combustion chamber. With an increase in the film hole size, compared to the 0.8 mm and 1 mm diameters, the heat flux curve becomes smoother and the values continuously increase. However, when the diameter is 1.2 mm, the heat flux curve shows little change. Therefore, as the film hole size increases, the wall temperature in the upstream region of the combustion chamber increases. However, as the size continues to increase, the temperature no longer rises and tends to stabilize.

## 7. Conclusions

This article employs numerical simulation methods to investigate the influence of different parameters on the film-cooling performance of a hydrogen–oxygen stoichiometric combustion chamber. Additionally, the study analyzes the differences in wall cooling between circular-hole-structured film and slot-structured film. Based on the analysis results, the following conclusions can be drawn:

1.  The EDC combustion model with finite-rate chemistry used in this study accurately predicts the flow and coupled heat transfer of a regenerative cooling combustion chamber. Experimental data quantitatively confirm the accuracy of calculating wall temperatures.
2.  The circular-hole-structured film is more prone to backflow and vortex formation in the upstream region of the combustion chamber, which carries away the cooling gas near the wall, leading to a decrease in cooling effectiveness. In contrast, the slot-structured film closely adheres to the inner wall of the combustion chamber, resulting in better cooling efficiency. Therefore, in the upstream region of the thrust chamber, the difference in maximum wall temperatures between the two structures is approximately 6%.
3.  With an increase in film hole size, for the circular-hole-structured film, cooling effectiveness improves in the vicinity of the panel area. However, in the downstream region of the combustion chamber, cooling effectiveness tends to deteriorate. For the slot-structured film, wall temperature in the upstream region of the combustion chamber also increases with significant changes in the circumferential direction. However, in the downstream region, wall temperature is almost unaffected by the film hole size.

**Author Contributions:** Methodology, H.R.; Resources, Z.L.; Writing–original draft, J.X. and Y.J. All authors have read and agreed to the published version of the manuscript.

**Funding:** This work is supported by the following: (1) Development of high-power-to-weight-ratio heavy-oil piston engine for general aviation at 150 (Project No.202201120401018); (2) Shanxi Province Applied Basic Research Program for Young Scientists: Research on Transcritical Combustion and Coupled Heat Transfer Characteristics of Pin-Injector in Supercritical Conditions (Project No.20210302124681); (3) Shanxi Province Applied Basic Research Program for Young Scientists (Project No.20210302124681, 20210302124385); (4) Shanxi Province Higher Education Science and Technology Innovation Project Funding (Project No.2021L069); and (5) Shanxi Province Science and Technology Major Project (Project No.202101120401007).

**Data Availability Statement:** Data is contained within the article.

**Conflicts of Interest:** Author He Ren was employed by the company Commercial Aircraft Corporation of China Ltd. The remaining authors declare that the research was conducted in the absence of any commercial or financial relationships that could be construed as a potential conflict of interest.

## Appendix A. Position of Circular Hole Structure Film

To investigate the influence of film-cooling hole position (step height) on wall-cooling performance, models with different film-cooling hole positions are designed based on the

circular hole structure film and simulated. As shown in Figure A1, schematic diagrams of wall temperature along the axial direction are presented for different step heights, where the initial model represents the model used in this paper's calculations, and the comparative model represents the model with film-cooling holes shifted downward (moved down by about 0.85D). It can be observed from the figure that the position of the film-cooling holes has a certain impact on the wall temperature with a slight increase in wall temperature as the film cooling hole position is shifted downward and a temperature increase of about 6%. Since such structures do not exist in engineering practice, this lack of practical significance led us to retain the steps according to engineering design when making comparisons in this paper. Comparisons were made with other engineering structures, which is more in line with practical considerations.

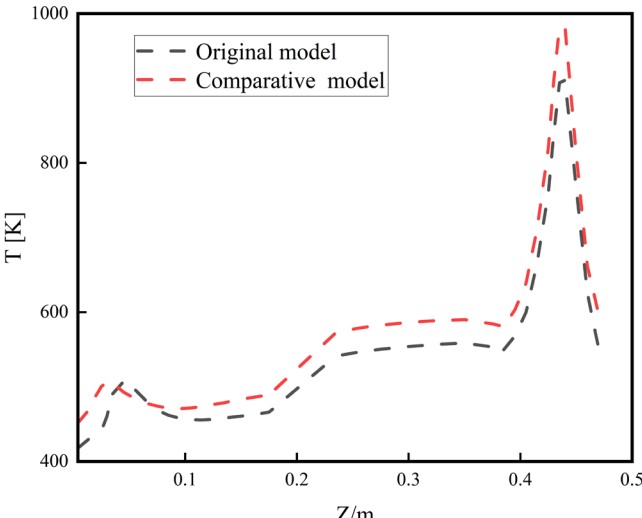

**Figure A1.** Schematic diagram of wall temperature along the axial direction.

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
