# Peer review of "Effects of Different Structural Film Cooling on Cooling Performance in a GO2/GH2 Subscale Thrust Chamber"

_aerospace, doi:10.3390/aerospace11060433_

Round 1
Reviewer 1 Report
Comments and Suggestions for Authors
The paper deals with the effects of circular hole and slot structure for investigating the cooling effectiveness with various parametric influence. The work is relevant in the area of rocket thrust chamber performance analysis. The problem statement is developed well and sufficient literature are referred to support the problem statement. Numerical studies have been carried out and validated properly. However, it lacks clarity in some areas and the paper can be accepted after incorporating minor revisions as mentioned below.
1. Clarity is needed in section 2.3 regarding the rationale for choosing the measurement points in temperature data
2.Again in section 2.3, what are the pipelines that are distributed radially? Clarity is needed here
3.Emphasis is required on the rationale of choosing mass flow rate as 2.4kg/s for coolant and cooling channel height as 2mm
4.Figure 5 does not have clarity in order to view symmetry and interface plane. Figure Colour figures would suffice.
5. Give reason for adopting case 2 grid for this study
6. Table the parameters of mass flow rate and hole dimensions for both circular hole and slot structure.
7. In figure 9, clarity is needed for model validation as it is vague in understanding which parameters have difference of 0.45 MW of power output between numerical and experimental calculation.
Comments on the Quality of English LanguageThe paper is written well and the structure of the paper is coherent and understandable. However to improve clarity, minor revisions are required in terms of syntax in the introduction section.
Author Response
Thank you for your letter and for the reviewers’ comments concerning our manuscript entitled “Effects of different structural film-cooling on cooling performance in a GO2/GH2 subscale thrust chamber”. Those comments are all valuable and very helpful for revising and improving our paper, as well as the important guiding significance to our research. We have studied comments carefully and have made corrections which we hope meet with approval.See the annex for details.

Reviewer 2 Report
Comments and Suggestions for Authors
This paper dwells on the distinct parameters that affect a hydrogen-oxygen stoichiometric combustion chamber's film cooling performance. Both experimental and numerical studies have been performed to investigate the effect of different parameters. Numerical simulations are performed using ANSYS Fluent. The specific comments and suggestions are as follows:
1. Please clearly label the important components of the thrust chamber (shown in Fig.2).
2. What type of thermocouples are used in this experimental measurement? Please add thermocouple uncertainty also.
3. What methods have been adopted for uncertainty estimation? The authors may add a separate table for uncertainty related to different measurements
4. The grids are not visible, additionally, some important information is missing such as grid uncertainty, grid independence study, and wall y+ (Y-plus) near important solid domains.
5. Pressure-based solvers are justified for sonic flow through the nozzle. Why density-based solver has not been used. Please add details of the turbulence model and convergence criteria.
6. The caption for Table 4 is incorrect. (page 8)
7. The authors may add a separate figure to explain the exact positions of thermocouples.
8. Figure 7 does not add value, authors may explain important terms related to numerical methodologies in the text.
9. Regarding Fig. 9, the numerical model underprediction at M=7.5%, however further increase of M to 12.5% the model overpredicts. Please add a physical reason behind this.
10. The conclusion section may be revised for better clarity.

Considerable for a scientific/technical presentation.
Author Response

(The authors gave the same response as above.)

Reviewer 3 Report
Comments and Suggestions for Authors
Author Response

(The authors gave the same response as above.)

Round 2
Reviewer 2 Report
Comments and Suggestions for Authors
The manuscript may be accepted as most of the important comments have been addressed.
Comments on the Quality of English LanguageNo specific comments on grammar, may check once.
Author Response
Thank you very much for your comments. We have done a check on the article and made corrections to the article.
Reviewer 3 Report
Comments and Suggestions for Authors
Thank you for corresponding to reviewer's comments. Please check the comments to each answer from the author.

Author Response
Thank you for your letter and for the reviewers’ comments . Those comments are all valuable and very helpful for revising and improving our paper, as well as the important guiding significance to our research. We have studied comments carefully and have made corrections which we hope meet with approval. The revisions are under the comments, marked in green.

Round 3
Reviewer 3 Report
Comments and Suggestions for Authors
Thank you for corresponding to reviewer’s comments.
(1) I confirmed your answer, therefore could you please add the text of comments No. 3, 6, 7 in the paper? Those information are important for us to understand precisely what you write. I also recommend to write the specific value of MR (fixed value in this research) if you can write.
(2) Regarding comment no.2, I still believe that the author's approach is not the fair comparison of the hole shape. The effect of step seems to affect to the result strongly (See Fig. 15). Could you please add the quantitative explanation that the effect of step doesn't affect to the result in this research ?
Author Response
Thank you for your suggestion, we have revised the original manuscript after careful consideration, and the reply draft is attached.

Round 4
Reviewer 3 Report
Comments and Suggestions for Authors
Thank you for corresponding to reviewer's comments.
1 comment : I confirmed your answer. Thank you for corresponding.
2 comment : Thank you for showing additional results. My opinion is that your conclusion is to compare the differences in wall cooling between circular-hole-structured film and slot-structured film, therefore those two effect should be fairly compared. If you want to focus on the engineering perspective, I have to point out to add the evaluation of the position of circular hole in the injector plate (but I don't want to expand to such additional perspective.). I recommend to add this quantitative evaluation including the figure in Appendix. in the paper that the effect of step is negligible in this research.
Author Response
Thank you for your feedback; we greatly appreciate your perspective and find it very insightful, and your input has inspired us. We have already incorporated the relevant content, which can now be found in Appendix A.